# Identification of human D lactate dehydrogenase deficiency

Glen R. Monroe [1,2], Albertien M. van Eerde[1,2], Federico Tessadori[1,2,3], Karen J. Duran[1,2],
Sanne M.C. Savelberg[1,2], Johanna C. van Alfen[4], Paulien A. Terhal[1], Saskia N. van der Crabben[5],
Klaske D. Lichtenbelt [1], Sabine A. Fuchs[5], Johan Gerrits[1], Markus J. van Roosmalen[1,2], Koen L. van Gassen[1],
Mirjam van Aalderen[1], Bart G. Koot[6], Marlies Oostendorp[7,8], Marinus Duran[9], Gepke Visser[5],
Tom J. de Koning[10], Francesco Calì[11], Paolo Bosco[11], Karin Geleijns[12], Monique G.M. de Sain-van der Velden[1],
Nine V. Knoers[1,2], Jeroen Bakkers [3,13], Nanda M. Verhoeven-Duif[1,2], Gijs van Haaften[1,2] & Judith J. Jans[1,2]

Phenotypic and biochemical categorization of humans with detrimental variants can provide valuable information on gene function. We illustrate this with the identification of two different homozygous variants resulting in enzymatic loss-of-function in *LDHD*, encoding lactate dehydrogenase D, in two unrelated patients with elevated D-lactate urinary excretion and plasma concentrations. We establish the role of LDHD by demonstrating that LDHD loss-of-function in zebrafish results in increased concentrations of D-lactate. D-lactate levels are rescued by wildtype *LDHD* but not by patients' variant *LDHD*, confirming these variants' loss-of-function effect. This work provides the first in vivo evidence that LDHD is responsible for human D-lactate metabolism. This broadens the differential diagnosis of D-lactic acidosis, an increasingly recognized complication of short bowel syndrome with unpredictable onset and severity. With the expanding incidence of intestinal resection for disease or obesity, the elucidation of this metabolic pathway may have relevance for those patients with D-lactic acidosis.

[1] Department of Genetics, University Medical Center Utrecht, Utrecht 3584 CX, The Netherlands. [2] Center for Molecular Medicine, University Medical Center Utrecht, Utrecht 3584 CX, The Netherlands. [3] Hubrecht Institute-KNAW and University Medical Center Utrecht, Utrecht 3584 CT, The Netherlands. [4] Bartiméus, Institute for the Visually Impaired, Doorn 3940 AB, The Netherlands. [5] Department of Metabolic Diseases, Wilhelmina Children's Hospital, University Medical Center Utrecht, Utrecht 3584 EA, The Netherlands. [6] Department of Pediatric Gastroenterology and Nutrition, Academic Medical Center, Amsterdam 1105 AZ, The Netherlands. [7] Department of Clinical Chemistry and Haematology, University Medical Center Utrecht, Utrecht 3584 CX, The Netherlands. [8] Laboratory of Clinical Chemistry, Deventer Hospital, Deventer 7416 SE, The Netherlands. [9] Laboratory Genetic Metabolic Diseases, Academic Medical Center, Amsterdam 1105 AZ, The Netherlands. [10] Section of Metabolic Diseases, Beatrix Children's Hospital, University Medical Center Groningen, Groningen 9713 GZ, The Netherlands. [11] Oasi Research Institute—IRCCS, Troina 94018, Italy. [12] Department of Child Neurology, Brain Center Rudolf Magnus, University Medical Center Utrecht, Utrecht 3584 CX, The Netherlands. [13] Department of Medical Physiology, University Medical Center Utrecht, Utrecht 3584 CX, The Netherlands. These authors contributed equally: Glen R. Monroe, Albertien M. van Eerde. These authors jointly supervised this work: Gijs van Haaften, Judith J. Jans. Correspondence and requests for materials should be addressed to G.v.H. (email: G.vanHaaften@umcutrecht.nl)

L actate exists in the human body as two optical isomers: L-lactate and D-lactate (Fig. 1). L-lactate is produced from pyruvate during anaerobic glycolysis and present in blood at concentrations 100 times greater than D-lactate; the use of a chiral derivatization reagent enables enantiomeric separation[1–3]. D-lactate is acquired exogenously by consumption of foods, by metabolism of chemicals such as propylene glycol, by intestinal bacteria production, or endogenously by methylglyoxal metabolism—a toxic product that is converted to D-lactate[4–8]. The methylglyoxal pathway is upregulated in various types of cancer as a consequence of the metabolic switch to aerobic glycolysis[9–11].

D-lactic acidosis develops when D-lactate levels in plasma are >3 mmol L$^{-1}$ and is a rare complication of short bowel syndrome[12]. Short bowel syndrome occurs after removal of a section of the small intestine due to malignancy or disease (i.e., Crohn's disease), or jejunoileal bypass surgery for obesity treatment[13–16]. The shortened small intestine impairs carbohydrate absorption and hence leads to increased carbohydrate delivery to colonic bacteria. As a consequence, colonic bacteria proliferate and foster an acidic environment favoring D-lactate producing species. Clinical features comprise primarily neurological symptoms. Management of D-lactate acidosis consists of restoring acid–base balance by bicarbonate infusion, and antibiotic treatment and carbohydrate restriction to reduce D-lactate producing bacteria[1]. The onset and degree of severity of D-lactic acidosis are still not well understood, nor are the pathways involved in D-lactate metabolism.

We report two patients with increased D-lactate excretion and elevated D-lactate concentration in plasma, yet without acidosis. Both patients have rare or novel variants in *LDHD* predicted to result in enzymatic impairment; metabolic studies expressing these variants in zebrafish and subsequent rescue by human wildtype LDHD establish the role of LDHD in D-lactate metabolism.

## Results

**Metabolic analysis reveals elevated D-lactate**. The first patient, born of Sicilian parents from the same village, was originally described by Duran et al.[17]. He was seen by a clinical geneticist at age 4 due to delayed motor and mental development. Urinary lactate excretion was high, and consisted almost exclusively of D-lactate instead of the common isomer L-lactate[17]. Upon reexamination at age 40 because of family counseling, he still had

an extremely increased D-lactate excretion (mean: 1686 mmol per mol creatinine) (Fig. 2a). Plasma analysis also revealed elevated D-lactate concentration (0.7 mM) (Fig. 2b). Furthermore, 2-hydroxyisovaleric acid and 2-hydroxyisocaproic acid were elevated in urine and plasma organic acid profiles. Upon identification of the increased D-lactate isomer, the chirality of these increased metabolites was subsequently also determined to be in the D-isomer form (Fig. 2c–f). Repeated antibiotic treatments failed to correct D-lactate levels, making a bacterial origin of this elevated metabolite unlikely. Additionally, the continued presence of increased D-lactate suggested that this was not a temporary metabolic disturbance. Array CGH analysis revealed a de novo 11p13 deletion, known to cause intellectual disability and explaining his syndromal features, though elevated D-lactate levels are not associated with this syndrome[18].

The unique biochemical phenotype of this patient prompted us to examine new patients for elevated D-lactate concentrations and we identified a second patient with a clinical diagnosis of West syndrome and elevated D-lactate in urine and plasma, as well as increased levels of both D-2-hydroxyacids (Fig. 2). As increased D-lactate excretion is not a known feature of either an 11p13 deletion or West syndrome, we investigated if this perturbation could be due to a different genetic cause, particularly as studies in neurometabolic patient cohorts have reported a high rate (13–14%) of patients with multiple molecular diagnoses[19,20].

**Identification of LDHD responsible for D-lactate metabolism**. As the parents of Patient 1 originated from the same Sicilian village, we hypothesized that they may share some degree of consanguinity. We therefore analyzed homozygosity regions identified by SNParray for genes which might relate to D-lactate excretion (Supplementary Table 1). The fifth largest stretch of homozygosity of 3.5 MB contained 28 protein coding genes including *LDHD* (Supplementary Table 2). LDHD has been identified as a putative lactate dehydrogenase in mammals but the function in humans has not been elucidated[21,22]. We performed Sanger sequencing of exonic regions and intronic/exonic boundaries of DNA isolated from blood of the patient and identified a homozygous *LDHD* nonsynonymous variant NM_153486.3:c.1388C>T [https://www.ncbi.nlm.nih.gov/nuccore/NM_153486.3], p.(Thr463Met) (Fig. 3a). Both parents were heterozygous carriers. The variant's predicted effect on protein function was classified as probably damaging (PolyPhen-2) and deleterious (Sorting Intolerant From Tolerant; SIFT), and resided in a region highly conserved across multiple species (Fig. 3b)[23,24]. The variant was not present in large human population variant frequency databases such as the Genome of the Netherlands, the Exome Variant Server, the 1000 Genomes or our in-house dataset, although it is present heterozygously in 27 individuals in the Genome Aggregation Database (gnomAD) and has recently been listed in dbSNP Build 151 as rs764877688[25–28]. Sanger sequencing of 200 additional individuals from the same region as that of our patient's parents was performed to investigate if the variant was present at a higher frequency in the Sicilian population but not represented in larger databases. However, no other individuals with this variant were detected.

Subsequent Sanger sequencing of *LDHD* in Patient 2 identified a novel homozygous variant at NM_153486.3:c.1122G>T [https://www.ncbi.nlm.nih.gov/nuccore/NM_153486.3], p.(Trp374Cys) that was not present in any large human population variant frequency database, predicted to be probably damaging and deleterious by PolyPhen-2 and SIFT, respectively[23,24], and resided within an area conserved amongst chordates (Fig. 3c, d). A Gene Matcher search failed to identify similar patients with *LDHD* variants (search performed August 13, 2018)[29]. Taken together,

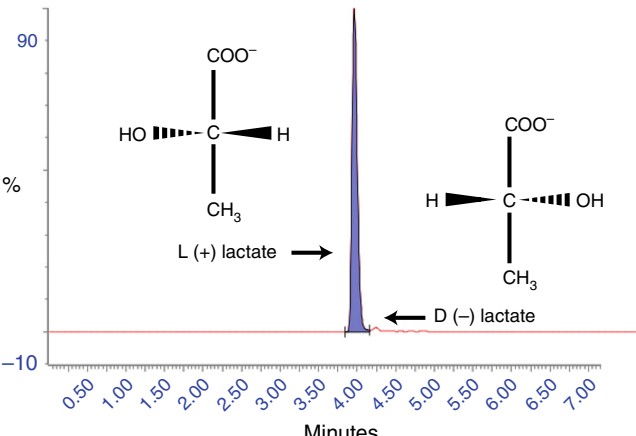

**Fig. 1** Lactate optical isomers. Mass-spectrometry chromatogram of the separation of L-lactate (purple) and its optical isomer D-lactate. Note the relative position of the hydroxyl group and hydrogen atom. L-lactate is present at approximately 100 times that of D-lactate in the plasma of a healthy control

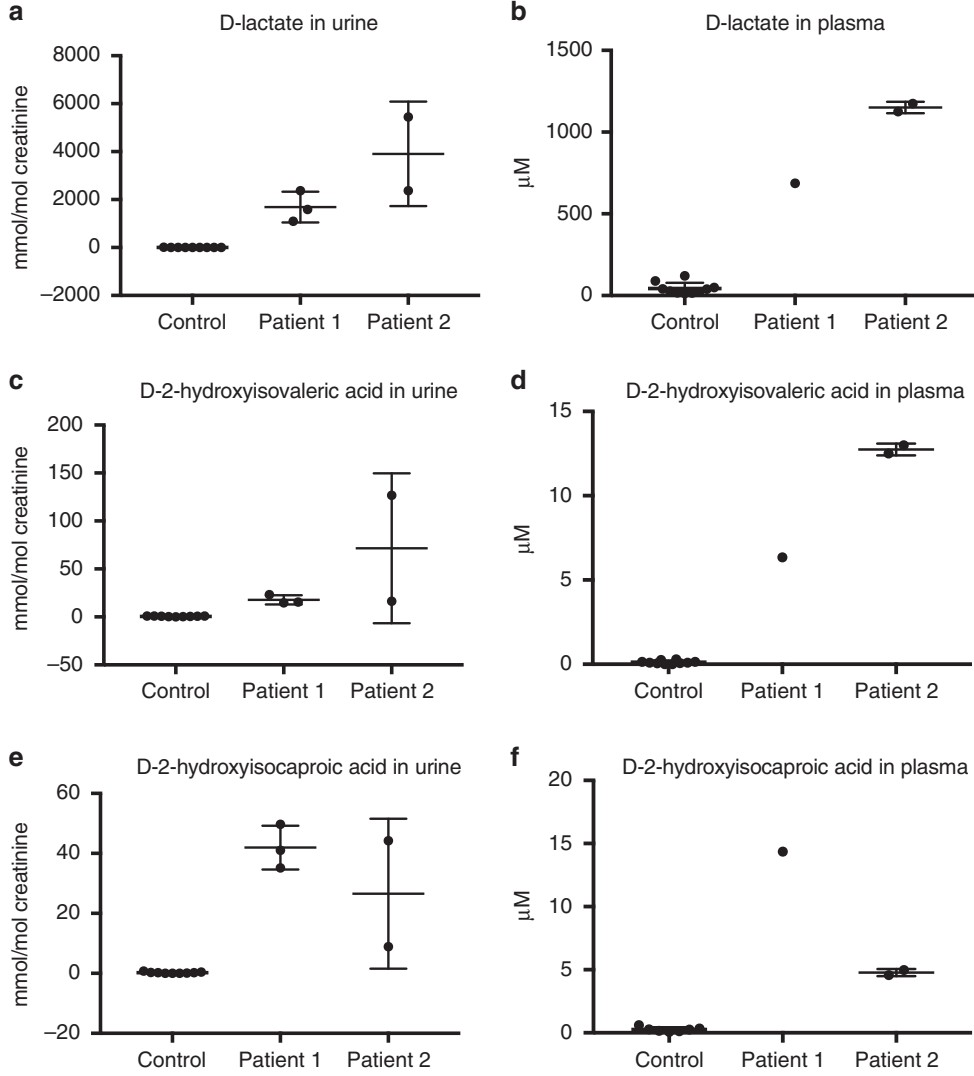

**Fig. 2** Metabolic analysis reveals elevated D-lactate of two patients. The levels of D-lactate in **a** urine and **b** plasma are elevated in our patients compared to controls. Increased levels of D-2-hydroxyisovaleric acid (**c**, **d**) and D-2-hydroxyisocaproic acid (**e**, **f**) are also observed in urine and plasma, respectively. Separate metabolic measurements for Patient 1 and 2 in urine ($n = 3$; $n = 2$) and plasma ($n = 1$; $n = 2$) are shown. Mean and standard deviation shown

we hypothesized that the *LDHD* variants observed in these patients resulted in LDHD dysfunction and elevated D-lactate plasma concentration and urinary excretion.

**Zebrafish LDHD knockout and human LDHD microinjection.** We utilized the zebrafish as a model organism to verify the effect of LDHD loss-of-function and establish the role of LDHD in D-lactate metabolism. We acquired the *ldhd*[sa15623] zebrafish line through the European Zebrafish Resource Center (KIT-EZRC, Karlsruhe, Germany). This line carries a disrupted essential donor splice site following exon 3 resulting in a premature stop codon (Allele sa15623; Zv9 chr25:g.13920446T>G)[30]. Since *ldhd*[sa15623] homozygous fish remain viable and fertile, maternal and zygotic (MZ) mutant embryos (*ldhd*[−/−]) were used for phenotypic and metabolic evaluation of LDHD deficiency effects. Phenotypically, 3 dpf (days post fertilization) *ldhd*[−/−] embryos showed no visible abnormalities compared to wildtype (Fig. 4a), and displayed no ocular abnormalities at 5 dpf (Supplementary Fig. 1). However, metabolic analysis revealed elevated levels of D-lactate, but not L-lactate, in *ldhd*[−/−] larvae compared to wildtype (Fig. 4b, c). This confirmed conservation of LDHD function in zebrafish and

demonstrated the biochemical phenotype of LDHD loss-of-function, without evident clinical phenotype.

To study the effect on LDHD function of the two separate missense variants present in our patients, we microinjected *LDHD* wildtype, *LDHD* variant Thr463Met or *LDHD* variant Trp374Cys human mRNA into 1-cell embryos of either wildtype or *ldhd*[−/−] zebrafish. 3 dpf embryos from the different conditions were pooled in separate groups of 10 embryos for D-lactate analysis. Wildtype zebrafish larvae showed no phenotype upon *LDHD* wildtype or variant injection. Additionally, physiological D-lactate levels in wildtype zebrafish were not altered by microinjections of wildtype *LDHD*, variant Thr463Met *LDHD*, or variant Trp374Cys *LDHD*, showing that endogenous LDHD is sufficient to reduce D-lactate levels below the detection level. However, *ldhd*[−/−] uninjected embryos showed a significant D-lactate increase that was restored to baseline levels by mRNA microinjection of the wildtype *LDHD* sequence. In contrast, microinjection of variant Thr463Met *LDHD* or Trp374Cys *LDHD* mRNA did not rescue the metabolic phenotype (Fig. 4b). Finally, levels of L-lactate were not significantly altered among the different conditions, establishing that LDHD is stereospecific for D-lactate only (Fig. 4c). This confirmed that LDHD is responsible

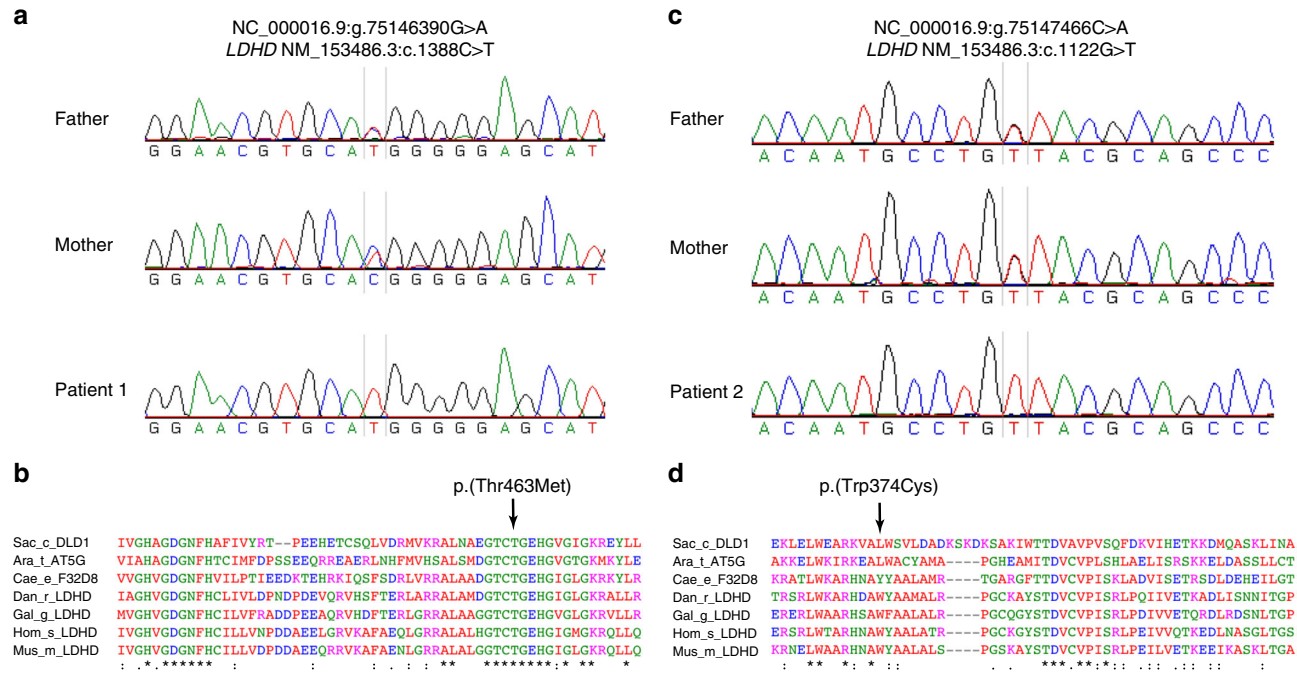

**Fig. 3** Identification of human loss-of-function variants in *LDHD*. **a** Sanger sequencing of Patient 1 identifies a *LDHD* homozygous nonsynonymous variant NM_153486.3:c.1388C>T, p.(Thr463Met). Both the mother and the father are heterozygous carriers of the variant. **b** The variant p.(Thr463Met) encodes for the amino acid methionine (M) instead of the normally present amino acid threonine (T) in a region that is highly conserved across multiple species. **c** Sanger sequencing of Patient 2 identifies a separate novel, homozygous nonsynonymous variant NM_153486.3:c.1122G>T, p.(Trp374Cys). Both the mother and the father are heterozygous carriers of the variant. **d** The variant p.(Trp374Cys) is in a region that is conserved across chordates. Sac_c_DLD1 Saccharomyces_cerevisiae_DLD1, Ara_t_AT5G Arabidopsis_thaliana_AT5G06580, Cae_e_F32D8 Caenorhabditis_elegans_F32D8.12, Gal_g_LDHD Gallus_gallus_LDHD, Hom_s_LDHD Homo_sapiens_LDHD, Mus_m_LDHD Mus_musculus_LDHD

for D-lactate metabolism, and that the homozygous variants present in our patients both result in decreased LDHD activity.

## Discussion

In humans, the ability to metabolize D-lactate to pyruvate has previously been attributed to D-2-hydroxy acid dehydrogenase. Mammals were thought to lack a functioning D-lactate dehydrogenase, though D-lactate dehydrogenases are present in yeast and prokaryotes[22,31–38]. D-lactate levels are not routinely studied, as analysis methods for plasma lactate are based on measuring the product of enzymatic conversion of L-lactate[34,39]. The detection of D-lactate thus requires specific assays based on bacterial LDHD or stereospecific mass spectrometry methods, both methods not routinely applied[40,41]. However, increased D-lactate can be responsible for an anion gap, as we see in our patients, and could potentially result in a greater risk of developing D-lactate acidosis. The first cases of D-lactic acidosis were identified in the late 1970s, with the majority of cases associated with short bowel syndrome[12,17,33]. While the work presented in this study does not justify *LDHD* screening in all patients with D-lactic acidosis, impairment of this metabolic pathway could conceivably result in an earlier onset or increased severity of acidosis. However, further research in this patient group is needed.

Neurologic symptoms commonly identified in D-lactic acidosis patients include altered mental status, slurred speech, ataxia, gate disturbance and less frequent manifestations ranging from aggressive behavior, hallucinations and paranoia to irritability, headache, and hunger[33]. The pathophysiological mechanism has not been clarified. As no clear correlation between D-lactate concentrations and the neurological phenotype has been demonstrated, it has been suggested that the neurological symptoms may be caused by other specific metabolic products of bacterial overgrowth, such as neurotoxin mercaptans, aldehydes, or others that may function as false neurotransmitters[1,4,13,33,42–44]. Patient 1 presented with intellectual disability and behavioral problems, which are both observed in 11p deletion syndrome[45]. One might speculate that the deficient D-lactate metabolism in Patient 1 contributed to the intellectual deficit or behavioral problems, and the neurological phenotype of Patient 2. However, given the phenotypic differences between our two patients, and the other genetic findings potentially explaining the rest of their phenotype, the loss-of-function LDHD phenotype in humans could represent a metabolic phenotype only. If LDHD loss-of-function is only associated with a biochemical phenotype, possibly this metabolic phenotype is overlooked or misclassified as L-lactic acidosis.

The diverse genetic findings in these patients highlight the need for deep phenotyping and comprehensive genomic analysis to detect both single nucleotide variants and copy number variants responsible for the patient phenotype. In the current study, this was accomplished utilizing a complementary approach of Sanger sequencing and WES to detect single nucleotide variants and array CGH/SNParray to identify large copy number variation (which WES is unable to perform). This is particularly necessary for those patients with rare complex phenotypes, where multiple genetic loci may contribute to the phenotype[19,20,46,47].

Increased levels of D-2-hydroxyisovaleric acid and D-2-hydroxyisocaproic acid were observed in both of our patients. The origin of these metabolites in vertebrate metabolism in unknown. Increased levels of the L-forms of these metabolites have been reported in the urine of patients with maple syrup urine disease (MSUD, MIM #248600)[48,49], lactic acidosis, ketoacidosis, as well as diabetic ketoacidosis patients[50,51]. The lactate dehydrogenase LDHA has experimentally been shown to be involved in the formation of the L-forms of these metabolites, particularly L-2-hydroxyisovaleric acid levels[52]. Heemskerk et al.

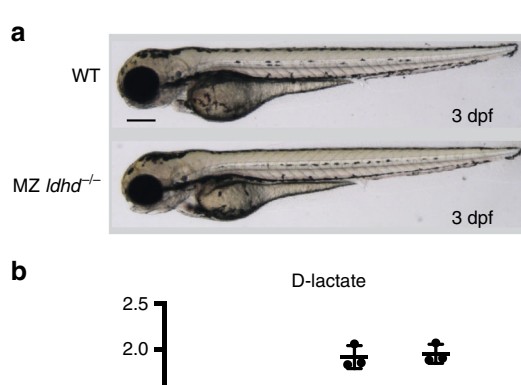

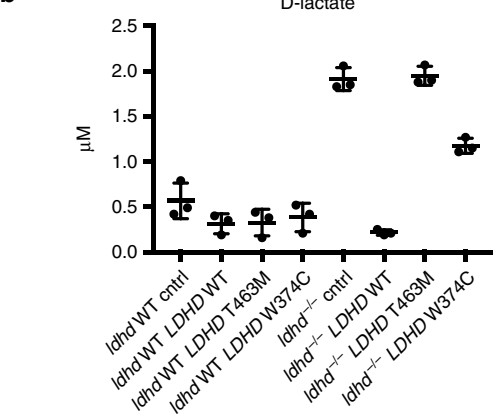

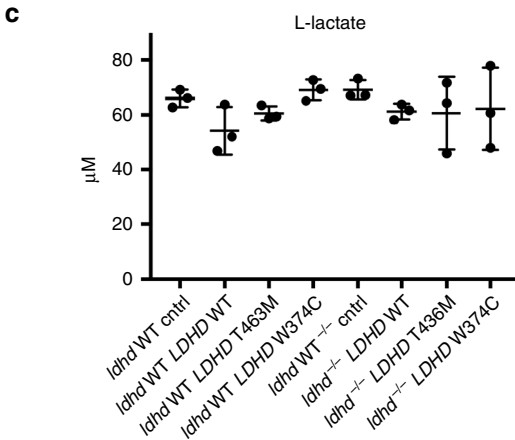

**Fig. 4** Zebrafish metabolic studies. **a** Maternal zygotic mutant *ldhd⁻/⁻* zebrafish larvae (3 dpf, lower panel) show no phenotype differences compared to wildtype zebrafish larvae (upper panel). Scale bar: 250 μm. Levels of D-lactate (**b**) and L-lactate (**c**) in response to LDHD activity. Wildtype zebrafish larvae (*ldhd* WT) and *ldhd⁻/⁻* zebrafish (*ldhd⁻/⁻*) at the 1 cell stage were subject to four conditions: uninjected (cntrl), microinjected with human wildtype *LDHD* RNA (*LDHD* WT), microinjected with patient variant Thr463Met *LDHD* RNA (*LDHD T463M*), or Trp374Cys *LDHD* RNA (*LDHD W374C*). Metabolic assays were then performed to detect D-lactate and L-lactate present at 3 dpf. The conditions where LDHD are nonfunctional result in higher levels of D-lactate; in contrast, no effect is seen in the levels of L-lactate. Measurements for each condition were performed on three batches of 10 embryos. Mean and standard deviation shown

show that LDHA can convert the ketoacid transamination products of valine, leucine, and isoleucine into L-2-hydroxyisovaleric acid, L-2-hydroxyisocaproic acid, and L-2-hydroxy-3-methylvaleric acid, respectively, and hypothesize that this function is necessary to prevent the accumulation of branched-chain ketoacids in hypoxic conditions. The increases of D-2-hydroxyisovaleric acid and D-2-hydroxyisocaproic acid that we have observed could suggest a role for LDHD in the metabolism of these D-isomer metabolites, but more research is needed to

clarify the mechanism of this little-known pathway. An intestinal bacterial origin of the D-2-branched-chain hydroxyacids cannot be ruled out completely. Spaapen et al. have demonstrated the presence of excessive amounts of D-2-hydroxyisocaproic acid in the urine of a patient with a short bowel syndrome, together with the typical D-lactate and, amongst others, the D-isomers of phenyllactic acid and 4-hydroxyphenyllactic acid which are metabolites of the amino acids phenylalanine and tyrosine, respectively[53]. Their findings suggest that the formation of D-2-hydroxyacids from parent amino acids is a common intestinal bacterial phenomenon. All thus formed D-2-hydroxyacids appear to be metabolized by endogenous LDHD, which has a limited overall capacity. In short bowel syndrome patients this limited capacity may be overwhelmed by excessive bacterial metabolism and result in accumulation of these D-2-branched-chain hydroxyacids. Similarly, in our patients, the accumulation of these hydroxyacids may be a result of LDHD loss-of-function.

In summary, we report two patients with different homozygous *LDHD* missense variants, both resulting in enzymatic loss-of-function leading to massive D-lactate excretion and increased D-lactate plasma concentrations. Zebrafish metabolic studies establish that LDHD loss results in increased D-lactate concentration that is rescued by wildtype *LDHD*, but not by patient variant *LDHD*. We therefore conclude that *LDHD* is essential for human D-lactate metabolism. As no other phenotype was observed in *ldhd⁻/⁻* zebrafish, it remains to be elucidated whether *LDHD* deficiency is a biochemical anomaly only, or if it contributes to a human clinical phenotype. Although plasma concentrations of D-lactate did not cause acidosis in our patients, individuals harboring *LDHD* variants that are detrimental to enzymatic function may be at risk to develop D-lactic acidosis due to short bowel resection or gastric bypass, a combination of factors that could overwhelm the body's ability to metabolize D-lactate. LDHD loss-of-function should thus be included in the differential diagnosis of D-lactate acidosis. Finally, this first in vivo identification of the role of *LDHD* in human D-lactate metabolism demonstrates the value of humans with deleterious variants in revealing and characterizing gene function.

## Methods

**Clinical patient reports**. Patient 1 was originally described 40 years ago as a novel case of D-lactic aciduria[17]. The patient was born of Sicilian parents, at the age of 1 year his mental and motor development were severely impaired. He was presented to the clinic at the age of 4 years. Briefly, he had been born of an uneventful pregnancy at 40 weeks, with microcephaly (OFC 34.5 cm), slanting of the eyelids, bilateral inguinal hernia, and aniridia. Development quotient at the age of 1 year was 51 at Griffiths scale and motor development was also severely impaired. At the age of 40, the family presented again for genetic counseling. By then he was known to carry a de novo deletion on chromosome 11p13. His medical history included a severe mental handicap with behavioral problems, cryptorchidism, blindness (aniridia, with later onset of cataract and glaucoma) and epilepsy until the age of 14. He had not developed a Wilms tumor. Upon reexamination at the age of 40, he was normocephalic (OFC 57 cm), with down slanting eyelids, a protruding lower lip, mildly dysplastic helices, and patches of greying hair. The second digits of his feet were longer than the halluces.

Patient 2 was the first child of consanguineous (first cousins) Moluccan parents (Indonesia), born after an uneventful pregnancy. He had a normal development until the age of 5 months when he developed seizures fitting West syndrome. He experienced developmental regression with severe hypotonia including headlag, slipping through and lost social interaction and remains developmentally delayed. Physical examination did not reveal other abnormalities or dysmorphic features. With antiepileptic drugs (Levetiracetam and Vigabatrin), seizure frequency decreased and interaction improved. His motor development has now significantly improved, but he remains developmentally delayed.

**Patient genetic investigations**. For Patient 1, clinical suspicion of 11p deletion syndrome resulted in array CGH analysis at the Genome Diagnostics Department of Genetics, University Medical Center Utrecht, The Netherlands. A paternal deletion 3q24 (260 kb) was detected in the patient, as well as a 11.13 Mb de novo deletion spanning from 11p14.1 to 11p12, confirming the 11p deletion syndrome. The patient's intellectual disability, ophthalmologic features, cryptorchidism, and

**Table 1 Optimized MRM settings**

| Component | Parent ion (m per z) | | Daughter ion (m per z) | |
|---|---|---|---|---|
| | Collision energy (eV) | | Dwell time (s) | |
| [$^{13}C_3$]-L-lactate | 307.95 | 91.95 | 8.0 | 0.1 |
| L/D-lactate | 304.95 | 88.95 | 8.0 | 0.1 |
| L/D-2-OH-isovaleric acid | 333.00 | 117.00 | 8.0 | 0.1 |
| L/D-2-OH-isocaproic acid | 347.00 | 131.00 | 8.0 | 0.1 |

seizures were consistent with 11p deletion syndrome—also known as WAGR syndrome (Wilms tumor; Aniridia; Genital anomalies; Retardation)—which was concluded to be likely causal for most of the patient's phenotype[18,54,55].

Diagnostic Sanger sequencing of *PDHX*, a gene in which homozygous or compound heterozygous variants are known to cause lactic acidemia (MIM #245349), revealed no causal variants (Department of Genetics, Genome Diagnostics, University Medical Center Groningen, The Netherlands). At the age of 40, a high-resolution SNParray was performed to establish the minimal breakpoints chr11:29651299 to chr11:40817882 of the de novo deletion (GRCh37/hg19 genome build). Moreover, several large regions of homozygosity were observed, confirming relatedness of the parents (Supplementary Table 1). The largest homozygous stretch contained an 11.1 MB region with 51 genes, encompassing the 11p13 region including *PAX6* and *WT1*. Genes present in the identified regions were annotated on function and literature consulted to establish if the gene could be a candidate for elevated D-lactate levels. Of the genes that were present in homozygous regions, *LDHD* was the most likely candidate as it had previously been identified as a putative D-lactate dehydrogenase (Supplementary Table 2).

Subsequently, the exonic and exonic/intronic boundaries of *LDHD* were screened by Sanger sequencing (primers available in Supplementary Table 3), identifying the variant NM_153486.3:c.1388C>T, p.(Thr463Met). To determine if this variant was population specific in the isolated Sicilian community of the family, 200 additional individuals were screened for variants using Sanger sequencing. One hundred of these individuals originated from within a 100 km radius of the town of patient's birth in the province of Palermo; a further 100 individuals were from neighboring Sicilian areas outside of the 100 km radius.

For Patient 2, whole exome sequencing was performed to attempt to find a genetic cause of West Syndrome, according to standard diagnostic procedures and WES quality criteria at UMC Utrecht, The Netherlands. Patient–parent trio whole exome sequencing in search for a genetic cause of the patient's seizures identified a de novo variant in *CACNA1B* (NM_000718.3: c.1429C>T, p.(Arg477Cys); Calcium channel, voltage dependent, N type, Alpha-1B subunit), a voltage-dependent $Ca^{2+}$ channel. Variants in *CACNA1B* have not previously been linked to West Syndrome, though a different heterozygous *CACNA1B* missense variant (NM_000718.3: c.4166G>A, p.(Arg1389His)) segregating in affected individuals of a Dutch family with autosomal dominant dystonia-23 (MIM: 614860) has recently been reported[56]. Additionally, heterozygous variants in paralogous *CACNA1A* are known to cause episodic ataxia, type 2 (MIM: 601011), as well as epileptic encephalopathy (MIM: 617106)[57]. Although functional work and identification of other patients with *CACNA1B* variants will be essential to establish a molecular diagnosis for this patient's seizures, we consider the variant a strong candidate to explain the epilepsy phenotype of the patient.

To determine if the patient's high D-lactate levels could be due to LDHD deficiency, a high-resolution SNParray was performed to identify if *LDHD* was present in the regions of homozygosity. The exonic and exonic/intronic boundaries of *LDHD* were subsequently screened by Sanger sequencing, identifying the variant NM_153486.3:c.1122G>T, p.(Trp374Cys).

Informed consent was obtained for both participants and all relevant ethical regulations were adhered to. Informed consent for whole-exome sequencing as a part of the diagnostic process (approved by the Medical Ethical Committee of the University Medical Center Utrecht) was obtained for Patient 2 and parents.

**Metabolic studies in zebrafish.** The ENU-mutagenized *LDHD* knockout zebrafish line *ldhd*$^{sa15623−/+}$ was obtained via the European Zebrafish Resource Center (EZRC; ZFIN ID: ZDB-GENE-030131-6140) at the Karlsruhe Institute of Technology (KIT) to evaluate the D-lactate levels in zebrafish larvae. The line contains the variant chr25:g.13920446T>G (Zv9 zebrafish genome build) that disrupts an essential splice site following exon 3, resulting in a knock-out of *LDHD* (http://www.sanger.ac.uk/sanger/Zebrafish_Zmpgene/ENSDARG00000038845#sa15623). The heterozygous carriers were grown to adults and incrossed to produce the *ldhd*$^{−/−}$ line, which is adult viable and fertile. Incrossing the *ldhd*$^{−/−}$ line provided the maternal and zygotic mutant embryos for this study. Animal experiments complied with all ethical regulations and were approved by the Animal Experimentation Committee of the Royal Netherlands Academy of Arts and Sciences.

Expression constructs were created using the wildtype human sequence of *LDHD* wildtype and the *LDHD* Thr463Met and Trp374Cys variants to evaluate if the D-lactate phenotype could be introduced and/or subsequently rescued in zebrafish larvae. Human *LDHD* cDNA (accession number NM_194436.1) in the entry vector pCMV-ENTRY (Origene, Rockville, MD, USA) was inserted into the expression vector pCS2+/GW (Thermo Fisher Scientific, Waltham, MA, USA). Site-directed mutagenesis was performed to introduce the Thr463Met variant by using primers 5-TCTCCACGGAACGTGCATGGGGGGAGCA-3 and 5-TGCTCCCCATGCACGTTCCGTGGAGA-3 or for the Trp374Cys variant by using primers 5-GGCACAATGCCTGTTACGCAGCCCTGG-3 and 5-CCAGGGCTGCGTAACAGGCATTGTGCC-3. These three plasmids were then linearized by NotI digestion and the reaction was cleaned up using the Qiaquick PCR purification kit (Qiagen, Hilden, Germany). RNA was synthesized from the linear DNA using the mMESSAGE mMACHINE SP6 transcription kit (Thermo Fisher Scientific, Waltham, MA, USA) and cleaned with the RNeasy mini kit (Qiagen, Hilden, Germany). Five ng/μl mRNA of the *LDHD* wildtype, Thr463Met *LDHD* variant, or Trp374Cys *LDHD* variant was used for microinjections into 1-cell stage *ldhd*$^{−/−}$ or wildtype zebrafish embryos. A control group of uninjected wildtype zebrafish and *ldhd*$^{−/−}$ zebrafish was also included for every experiment. At 3 dpf, three batches of 10 embryos were collected for each condition, pooled and stored at −80 °C.

For the zebrafish metabolic measurements, 200 μL methanol and 25 μL of internal standard solution (containing 434.75 μmol L$^{−1}$ [$^{13}C_3$]-L-lactate) was added to the separate three batches for each of the eight conditions. Next, sample extracts were homogenized with 0.5 mm zirconium oxide beads (product #ZrOB05, Next Advance, Inc., Averill Park, NY) using a bullet blender (model BBX24B-CE, Next Advance, Inc., Averill Park, NY) for 2 × 5 min, speed 8 at 4 °C. The extracts were centrifuged at 4 °C and 15,871 × g for 5 min. The supernatant was pipetted into a reaction vial and evaporated completely under a gentle stream of nitrogen at a temperature of 50 °C. Fifty μL of freshly made DATAN was added. The vial was capped, vortexed, and heated at 75 °C for 30 min. After 30 min, the vial was allowed to cool down to room temperature, and the mixture was evaporated completely with a gentle stream of nitrogen. The derivatized residue was reconstituted with 150 μL Solvent A and chromatography and mass spectrometry were performed as described above.

To visualize zebrafish sections by hematoxylin and eosin (H&E) staining, wildtype and MZ *ldhd*$^{−/−}$ 5 dpf embryos were fixed in 4% paraformaldehyde in PBS at 4 °C overnight, washed in PBS-Triton X100 (0.1%v/v), embedded in paraffin sectioned transversally at 10 μm intervals. Sections were subsequently stained with hematoxylin and eosin according to a standard protocol.

**Mass spectrometry methods.** Chemicals and reagents were obtained from various suppliers. Sodium L (+) lactate, sodium D (−) lactate, D-2-hydroxyisovaleric acid, L-2-hydroxyisovaleric acid, L-2-hydroxyisocaproic acid, (+)-O,O-diacetyl-L-tartaric anhydride (DATAN), and ammonium formate were obtained from Sigma Aldrich (St. Louis, MO, USA). [$^{13}C_3$]-Sodium L (+) lactate was obtained from Cambridge Isotope Laboratories (Tewksbury, MA, USA). Dichloromethane and acetic acid anhydrous were obtained from Merck (Kenilworth, NJ, USA). Acetonitrile, methanol, and formic acid (all at ULS/MS grade) were obtained from Biosolve (Dieuze, France). D-2-hydroxyisocaproic acid was obtained from Bachem AG (Bubendorf, Switzerland).

For plasma sample preparation, 25 μL of plasma was added to 25 μL of internal standard solution (containing 434.75 μmol L$^{−1}$ [$^{13}C_3$]-L-lactate). Samples were mixed thoroughly and subsequently deproteinized with a 600 μL mixture of methanol:acetonitrile (1:1, by volume) and centrifuged at 4 °C for 5 min at 15,871 × g. The supernatant was pipetted into a reaction vial and evaporated completely under a gentle stream of nitrogen at a temperature of 50 °C. Fifty μL of freshly made DATAN (50 mg mL$^{−1}$ dichloromethane: acetic acid (4:1, by volume)) was added. The vial was capped, vortexed, and heated at 75 °C for 30 min. After 30 min, the vial was allowed to cool down to room temperature, and the mixture was evaporated completely with a gentle stream of nitrogen. The derivatized residue was reconstituted with 150 μL Solvent A (1.5 mM ammonium formate (pH = 3.6)).

For urine sample preparation, 25 μL of internal standard solution, 25 μL urine, and 300 μL of methanol were pipetted into a reaction vial. Samples were mixed thoroughly and evaporated completely under a gentle stream of nitrogen at a temperature of 50 °C. Fifty μL of freshly made DATAN was added. The vial was capped, vortexed, and heated at 75 °C for 30 min. After 30 min, the vial was allowed to cool down to room temperature, and the mixture was evaporated completely with a gentle stream of nitrogen. The derivatized residue was reconstituted with 300 μL Solvent A.

Samples were analyzed by reversed phase LC-tandem MS using an Acquity UPLC BEH C18 analytical column (100 × 2.1 mm, 1.7 μm; Waters, Milford, MA, USA). Detection was carried out using a Xevo TQ tandem mass spectrometer (Waters, Milford, MA, USA), which was operated in negative multiple-reaction-monitoring (MRM) mode. UPLC analysis was performed using a binary gradient at a flow of 0.5 mL min$^{−1}$ using an Acquity UPLC (Waters, Milford, MA, USA). Solvent A was 1.5 mM ammonium formate (pH = 3.6), and solvent B was acetonitrile. A linear gradient was started at 0.5% B, and ramped to 3% B in 3 min, further ramped to 40% in 2 min, held at 40% B for 2 min and returned to 0.5% B in 1 min. The column was equilibrated for 1 min at the initial composition. Injection volume was 10 μL, and column temperature was set at 40 °C. Samples were kept at 6 °C. Chromatograms were acquired and processed with Masslynx V4.1 SCN 843 (Waters, Milford, MA, USA).

Optimal conditions for all parents were found at a capillary voltage of 1.5 kV and a cone voltage of 10 V. The source and desolvation temperature were 150 and 600 °C, respectively. The cone gas flow and desolvation gas flow were 0 and 800 L h$^{-1}$, respectively. To establish the most sensitive daughter ions, the collision energy was set at 8 eV with a collision gas flow of 0.15 mL min$^{-1}$ (Table 1).

Calibration standard curves for all compounds were made in Milli-Q water. Calibration curves were obtained by linear regression of a plot of the analyte concentration versus the peak-area ratio of the analyte/internal standard area. For all the analytes, [13C$_3$]-L-lactate was used as an internal standard.

## Data availability

The WES dataset for Patient 2 was generated during patient care. The family consented to deposition of data solely pertaining to this study, therefore the dataset has not been made publicly available, but is available from the corresponding author upon reasonable request. The accession codes of the two LDHD variants uploaded to ClinVar are SCV000840083 and SCV000840084.

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

## Acknowledgements

The authors would like to thank the patients and their families for their participation in this study. The authors also wish to thank Dr. Mies M. van Genderen, the patient's ophthalmologist at Bartimeus, Institute for the Visually Impaired, Doorn, The Netherlands. Finally, thank you to Jeroen Korving (Hubrecht Institute-KNAW) for providing the zebrafish histology staining. A.M.v.E. is supported by The Dutch Kidney Foundation (KSTP12-010).

## Author contributions

A.M.v.E., G.v.H. and J.J.J. conceived the study and G.v.H. and J.J.J. supervised the project. G.R.M. wrote the paper and assembled the data. F.T. and S.M.C.S. designed and performed the zebrafish expression experiments. K.J.D. created the LDHD expression constructs. A.M.v.E., J.C.v.A., P.A.T., S.N.v.d.C., K.D.L., S.A.F., K.L.v.G., M.v.A., B.G.K., M.O., M.D., G.V., T.J.d.K., K.G., M.G.M.d.S.-v.d.V. contributed patient clinical, metabolic, or genetic data. J.G. performed metabolic measurements. M.J.v.R. created bioinformatic scripts for data analysis. F.C. and P.B. contributed DNA for the Sicilian population sequencing. N.V.K., J.B. and N.M.V.-D. supervised the study. All authors discussed and commented on the manuscript.

## Additional information

**Competing interests:** The authors declare no competing interests.

