## [Peer review file · Nature Communications]

Reviewer #1 (Remarks to the Author):

In this elegant study Dr. Monroe and coworkers shed light on the physiologic function of D-lactate dehydrogenase (LDHD) and establish its role in mammalian metabolism. In healthy man D-lactate is present in small amounts which can be handled by the low activity of LDHD. Previously only D-2-hydroxy acid dehydrogenase was thought to metabolize D-lactate. Under certain clinical conditions (e.g. short bowel syndrome [SBS] or post bariatric surgery) large amounts of D-lactate enter the circulation from the colon where D-lactate producing bacteria feast on abundant carbohydrates delivered to the colon (D-lactate may also have other sources such as methylglyoxal metabolism). The presence of a genetic variant that causes LDHD deficiency, as convincingly shown by the authors, would be detrimental if D-lactic acidosis would develop due to the mentioned causes such as SBS. Such a coincidence may indeed be very rare and the authors' suggestion to test for LDHD deficiency before bariatric surgery needs to be seriously questioned. Gastric bypass for the treatment of morbid obesity is now frequently applied in western countries, i.e. every year 1 in 2000 persons gets it or 500 operations are performed per year per 1 million population. The prevalence of LDHD deficiency is not known, maybe the authors could make an educated guess and calculate how often one would find it in patients before bariatric surgery. Maybe the costs of analysis would be prohibitive when applying usual health care economics. For sure surgeons who perform bariatric surgery, gastroenterologists and intensive care physicians should know about genetically determined LDHD deficiency and if appropriate should initiate testing for it in patients with D-lactic acidosis and acute metabolic encephalopathy. The average internist still knows little or nothing about D-lactic acidosis. The authors now show convincingly that genetic variants with decreased or absent LDHD activity can lead to higher serum levels of D-lactate and to D-lactic aciduria.

Reviewer #2 (Remarks to the Author):

In the manuscript entitled "Identification of human D-lactate dehydrogenase deficiency" Monroe and colleagues describe two patients with elevated D-lactic acid levels that were found to have different mutations in the LDHD gene, leading to elevated levels of D-lactic acid in urine and plasma. Computational analysis predicts loss of function, and zebrafish *ldhd* mutants exhibit elevated D-lactic acid levels that cannot be rescued with the human mutant alleles.

Overall, this is an interesting manuscript that identifies the likely causative gene for the biochemical abnormalities observed in these patients. I have two comments:

- The authors need to pay more attention to the presentation of the data and the flow of the manuscript. For example, while the individual write-up of the patients' history is detailed, the manuscript itself does not provide enough detail, so that the reference to sequencing another 200 patients from Sicily is without context. Similarly, the reasoning why D-lactic metabolites are being assayed (as shown in Figure 2 C-E) only makes sense after reading the discussion. Also, please let genomic/gene location for the graph in 3A+C, and add the *Danio rerio* genetic sequence here. The authors should try to make the manuscript more reader-friendly.
- It would be interesting to know whether there are any organ defects in the zebrafish mutants, which could be easily assayed for eye phenotypes (such as aniridia) or other organ development. That is a straight-forward assessment.

Reviewer #3 (Remarks to the Author):

Monroe and co-authors report two unrelated cases with a deficiency of human D-lactate dehydrogenase deficiency.

Overall the manuscript is well-written with solid experimental data validating the detrimental effect of the LDHD variants on protein function causing an elevation of D-lactate in humans and zebrafish.

The authors' conclusion that this work provides the first in vivo evidence that LDHD is responsible for human D-lactate metabolism is justified; their findings are novel and this report is a valuable

addition to the literature.

There are several concerns however:

1) Major: The risk of metabolic decompensation of patients with LDHD deficiency during GI surgery / short bowel syndrome is highly speculative. It seems the authors have added this part to make their findings more clinically relevant. In my opinion it would be worthwhile to perform Sanger analyses of this gene in patients with short bowel syndrome and a confirmed episode of D-lactic acidosis; these data will (or will not) underpin the authors' claim. Without such data genetic screening is recommended in a group of patients without any evidence and this certainly does not favor efficient evidence-based medicine.

2) The authors mention several times that D-lactate acidosis has been described in patients with neurologic features in the literature; what are these features and have they carefully examined the LDHD knockout zebrafish for any subtle abnormalities (such as provoked seizures, brain morphologic findings, swim patterns etc) at a later age (ie beyond the larval stage)? Are the authors convinced that none of the neurologic findings in their patients is caused by the LDHD deficiency? Have they posted this gene on Gene Matcher to find more patients?

3) It is interesting that at least in 1 of their patients, authors have identified a pathogenic CNV as well as an AR condition and in the other possibly 1 AD condition and 1 AR condition. Such blended phenotypes arising from 2 distinct genetic aberrations in 1 individual have been reported in up to 5-10% of patients with rare phenotypes. In my opinion this finding deserves more attention and discussion, emphasizing the need for deep phenotyping and comprehensive exome analysis.

4) Should we be measuring D-lactate and L-Lactate more frequently? if so in what type of patients?

Reviewer #1 (Remarks to the Author):

In this elegant study Dr. Monroe and coworkers shed light on the physiologic function of D-lactate dehydrogenase (LDHD) and establish its role in mammalian metabolism. In healthy man D-lactate is present in small amounts which can be handled by the low activity of LDHD. Previously only D-2-hydroxy acid dehydrogenase was thought to metabolize D-lactate. Under certain clinical conditions (e.g. short bowel syndrome [SBS] or post bariatric surgery) large amounts of D-lactate enter the circulation from the colon where D-lactate producing bacteria feast on abundant carbohydrates delivered to the colon (D-lactate may also have other sources such as methylglyoxal metabolism). The presence of a genetic variant that causes LDHD deficiency, as convincingly shown by the authors, would be detrimental if D-lactic acidosis would develop due to the mentioned causes such as SBS. **Such a coincidence may indeed be very rare and the authors' suggestion to test for LDHD deficiency before bariatric surgery needs to be seriously questioned.** Gastric bypass for the treatment of morbid obesity is now frequently applied in western countries, i.e. every year 1 in 2000 persons gets it or 500 operations are performed per year per 1 million population. **The prevalence of LDHD deficiency is not known, maybe the authors could make an educated guess and calculate how often one would find it in patients before bariatric surgery. Maybe the costs of analysis would be prohibitive when applying usual health care economics.** For sure surgeons who perform bariatric surgery, gastroenterologists and intensive care physicians should know about genetically determined LDHD deficiency and if appropriate should initiate testing for it in patients with D-lactic acidosis and acute metabolic encephalopathy. The average internist still knows little or nothing about D-lactic acidosis. The authors now show convincingly that genetic variants with decreased or absent LDHD activity can lead to higher serum levels of D-lactate and to D-lactic aciduria.

Dear Reviewer, thank you for your comments. Our current study indeed elucidates the role of LDHD in D-lactate metabolism, which could have an impact on the severity or onset of D-lactic acidosis in those patients whom ultimately develop short bowel syndrome. We agree with your comments and those of Reviewer 3 in which screening for LDHD deficiency prior to bariatric surgery a) may not be completely warranted, given the evidence provided within the scope of this study alone, and b) would most likely be cost-prohibitive if a thorough cost analysis calculation was undertaken.

As the reviewer notes, in 2017 gastric bypass surgery was applied in approximately 1 in 1428 persons/year in the USA to anywhere from 1 in 13888 persons/year (Germany) to 1077 persons/year (Belgium) in Europe^{1,2}. The prevalence of LDHD deficiency is not known and difficult to calculate. In ExAC, a database containing exome sequences of 60,706 unrelated individuals, there are 48 loss-of-function alleles (stop gained, splice acceptor, frameshift). This results in an estimate of LDHD deficiency in 1 in 6.4 million individuals (48 LOF alleles in a total of 121412 alleles; $(48/121412) * (48/121412) = 0.0000001563$ or 1.56 individuals with two loss-of-function alleles in 10 million). This estimate is conservative because we cannot assess the impact of the missense variants, which, as functionally demonstrated in the patients within this manuscript, can also result in impairment of function. Admittedly, even if many missense variants do result in loss-of-function, the incidence of LDHD deficiency is still quite rare, arguing against systematic screening prior to surgery from either a health or a cost viewpoint.

In light of these numbers and the reviewer's comments we have modified the manuscript to solely present the data identifying LDHD as an essential enzyme for D-lactate metabolism, without recommending screening prior to surgery. Specifically, we have replaced the sentence in the abstract of: *"LDHD deficiency screening prior to surgery might be useful in*

preventing acidotic episodes or ensuring rapid treatment” in the abstract with “...the elucidation of this metabolic pathway may have relevance for those patients with D-lactic acidosis.” Additionally, we have removed the following sentence in the discussion: “Therefore, screening for LDHD deficiency before intestinal surgery may be useful for acidosis prevention in this patient group.” As we believe that the short bowel patient population would benefit from future follow-up studies investigating LDHD deficiency as a contributor to the severity or onset of D-lactic acidosis, we have left in the overview of this medical condition and suggest that LDHD analysis may be informative in patients with a severe phenotype.

Reviewer #2 (Remarks to the Author):

In the manuscript entitled “Identification of human D-lactate dehydrogenase deficiency” Monroe and colleagues describe two patients with elevated D-lactic acid levels that were found to have different mutations in the LDHD gene, leading to elevated levels of D-lactic acid in urine and plasma. Computational analysis predicts loss of function, and zebrafish *ldhd* mutants exhibit elevated D-lactic acid levels that cannot be rescued with the human mutant alleles.

Overall, this is an interesting manuscript that identifies the likely causative gene for the biochemical abnormalities observed in these patients. I have two comments:

The authors need to pay more attention to the presentation of the data and the flow of the manuscript. For example, while the individual write-up of the patients’ history is detailed, the manuscript itself does not provide enough detail, so that the reference to sequencing another 200 patients from Sicily is without context. Similarly, the reasoning why D-lactic metabolites are being assayed (as shown in Figure 2 C-E) only makes sense after reading the discussion. Also, please let genomic/gene location for the graph in 3A+C, and add the *Danio rerio* genetic sequence here. The authors should try to make the manuscript more reader-friendly.

Dear Reviewer, thank you for your response. We have critically gone through the manuscript to improve its flow and coherence and believe that the manuscript is significantly improved while retaining the critical information and adhering to the short format. We have specifically addressed your two points above, namely that of the Sicilian cohort Sanger sequencing and reason for why D-lactic metabolites are assayed, by providing more detail of both patients’ history in the main manuscript and describing in detail why the other D-metabolites were assayed. This was initially because they were observed to be elevated in Patient 1’s organic acid profile measurements. The text now reads:

- The first patient, *born of Sicilian parents from the same village*, was originally described by Duran et al.
- *As the parents of Patient 1 originated from the same Sicilian village, we hypothesized that they may share some degree of consanguinity.*
- Furthermore, 2-hydroxyisovaleric acid and 2-hydroxyisocaproic acid were *elevated* in urine and plasma *organic acid profiles*. *Upon identification of the increased D-lactate isomer, the chirality of these increased metabolites was subsequently also determined to be in the D-isomer form* (Figures 2c-f).

Additionally, we have included the HVGS notation of the variant's genomic/gene location in patient 1 and 2 in Figure 3A and 3C, respectively, and added the corresponding *Danio rerio* protein sequence in Figure 3B and 3D as well as the notation of the variants' effect at the amino acid level.

It would be interesting to know whether there are any organ defects in the zebrafish mutants, which could be easily assayed for eye phenotypes (such as aniridia) or other organ development. That is a straight-forward assessment.

We agree with the reviewer that similar organ defects in the Zebrafish compared to those observed in the patients could implicate deleterious *LDHD* variants to contribute to more systemic phenotypes. However, the eye phenotype present in patient 1 (namely blindness due to aniridia and onset of cataract and glaucoma) can be fully explained by deletion of 11p13 constituting WAGR syndrome. This locus contains *PAX6*, in which deleterious variants and deletions of this gene (or the larger gene region) result in aniridia, either isolated or in combination with syndromal features (WAGR syndrome)³. Additionally, homozygous mice *PAX6* knockout and zebrafish *PAX6* knockouts exhibit eye abnormalities, making the genetic evidence for this phenotype fully encompassed by the 11p13 deletion⁴⁻⁶. However, as the reviewer suggests this is a straight-forward assessment and would also be informative for other readers' interest. We have now included a hematoxylin and eosin staining of wildtype and maternal zygotic *ldhd*^{-/-} embryo eyes and assessed if there were any phenotype differences, including this information in the manuscript text and Supplementary Fig. 1.

The amended text now reads:

- “Phenotypically, 3dpf (days post fertilization) *ldhd*^{-/-} embryos showed no visible abnormalities compared to wildtype (Figure 4a), and displayed no ocular abnormalities at 5dpf (Supplemental Fig. 1).”

Reviewer #3 (Remarks to the Author):

Monroe and co-authors report two unrelated cases with a deficiency of human D-lactate dehydrogenase deficiency. Overall the manuscript is well-written with solid experimental data validating the detrimental effect of the LDHD variants on protein function causing an elevation of D-lactate in humans and zebrafish. The authors' conclusion that this work provides the first *in vivo* evidence that LDHD is responsible for human D-lactate metabolism is justified; their findings are novel and this report is a valuable addition to the literature.

There are several concerns however:

- 1) Major: The risk of metabolic decompensation of patients with LDHD deficiency during GI surgery / short bowel syndrome is highly speculative. It seems the authors have added this part to make their findings more clinically relevant. In my opinion it would be worthwhile to perform Sanger analyses of this gene in patients with short bowel syndrome and a confirmed episode of D-lactic acidosis; these data will (or will not) underpin the authors' claim. Without such data genetic screening is recommended in a group of patients without any evidence and this certainly does not favor efficient evidence-based medicine.

We agree with Reviewer #3 and Reviewer #1 that in retrospect this suggestion is too speculative to be included in the manuscript without evidence of screening in a cohort of patients with D-lactic acidosis and identifying *LDHD* deleterious variants. This follow-up research is very interesting but the recruitment and ethical permission to sequence a population of short bowel syndrome patients has been impossible to be performed in the timeline for this resubmission. In light of the reviewer comments we have removed every suggestion of preemptive screening before GI surgery as well as following short bowel syndrome from the manuscript.

We do believe that the impairment of this pathway could result in more severe or more frequent episodes of this little understood condition, and thus suggest that *LDHD* deficiency in these patients could be possible and assayed if the cause remains unexplained. Thus, we have rewritten the text to suggest that this approach would be interesting for future studies. We specifically now state in the Discussion: *"While the work presented in this study does not justify LDHD screening in all patients with D-lactic acidosis, impairment of this metabolic pathway could conceivably result in an earlier onset or increased severity of acidosis. However, further research in this patient group is needed."*

- 2) The authors mention several times that D-lactate acidosis has been described in patients with neurologic features in the literature.
 - a. What are these features?

We have now added the neurological features in the Discussion section when discussing the neurological phenotypes of our patients. Specifically, the text addition reads *"Neurologic symptoms commonly identified in D-lactic acidosis patients include altered mental status, slurred speech, ataxia, gait disturbance and less frequent manifestations ranging from aggressive behavior, hallucinations and paranoia to irritability, headache, and hunger⁷."*

- b. Have they carefully examined the LDHD knockout zebrafish for any subtle abnormalities (such as provoked seizures, brain morphologic findings, swim patterns etc.) at a later age (i.e. beyond the larval stage)?

As the contribution of D-lactate to the neurological phenotype of D-lactic acidosis is not known, and Patient 1 and Patient 2's intellectual disability or seizures are most likely due to their primary diagnosis of WAGR and West syndrome, respectively, we have not systematically analyzed the zebrafish in behavioral studies. Regardless, this is an intriguing question and one we have considered before the initial submission of this manuscript. The epilepsy of Patient 2 and also the altered mental status observed in D-lactic acidosis would in theory be apparent in adult zebrafish studies, as the reviewer suggests, by provoking epilepsy or analyzing swim patterns. However, there are several lines of reasoning arguing against this. In establishing this line, we have observed no seizures in homozygous embryos up until the developmental period for which we have ethical approval to observe the behavior of zebrafish (5 dpf). Reported zebrafish lines with metabolic defects are known to develop epilepsy beyond the larval stage; however, these lines are also prone to severe lethality⁸. Our *ldhd*^{-/-} line exhibits no indication of the presence of seizures and has been successfully in-crossed three times with no lethality. With these line characteristics, and the fact that both patients' neurological phenotype is explained by their primary genetic diagnosis, we cannot in good conscience request ethical permission for experiments beyond the larval stage for outcomes in which we expect a negative outcome – namely no phenotype.

- c. Are the authors convinced that none of the neurologic findings in their patients is caused by the LDHD deficiency?

We are unable to make a definite conclusion on the contribution of LDHD deficiency to the neurological phenotype of our patients. While increased D-lactate levels have been shown to be responsible for the acidosis, the role of D-lactate in metabolic encephalopathy is much less clearly established. There is no observed correlation between D-lactate levels and a clinical neurological phenotype and healthy patients receiving an infusion of D-lactate demonstrate no neurological symptoms. Additionally, Patient 1's intellectual disability and behavioural problems are also commonly reported as a characteristic of WAGR syndrome. We have added a discussion on D-lactate's contribution to the neurological status of patients, though we believe that both patients' neurological phenotypes are a result of their primary diagnosis of WAGR syndrome or West syndrome, respectively.

- d. Have they posted this gene on Gene Matcher to find more patients?

We have continually searched for new patients with *LDHD* variants, and unfortunately still, as of August 13, 2018, we have not identified any other patients, through Gene Matcher or other associated databases. We have added the following text into the manuscript: "A Gene Matcher search failed to identify similar patients with *LDHD* variants (search performed August 13, 2018)."

- 3) It is interesting that at least in 1 of their patients, authors have identified a pathogenic CNV as well as an AR condition and in the other possibly 1 AD condition and 1 AR condition. Such blended phenotypes arising from 2 distinct genetic aberrations in 1 individual have been reported in up to 5-10% of patients with rare phenotypes. In my opinion this finding deserves more attention and discussion, emphasizing the need for deep phenotyping and comprehensive exome analysis.

We agree with the reviewer that the contribution of two or more genetic causes to a blended or composite phenotype is increasingly identified in rare diseases, particularly in neurometabolic

disorders. We have drawn attention to this point with the below text in the Results section as well as with the below text in the Discussion section.

- (Results): As increased D-lactate excretion is not a known feature of either of these two syndromes, we investigated if this perturbation could be due to a different genetic cause, *particularly as studies in neurometabolic patient cohorts have reported a high rate (13-14%) of patients with multiple molecular diagnoses^{9,10}.*
- (Discussion): *The dual genetic findings in both of these patients highlights the need for thorough exome investigation and deep phenotyping in patients, particularly those with rare complex phenotypes, to uncover all possible genetic contribution to the phenotype.*

4) Should we be measuring D-lactate and L-Lactate more frequently? if so in what type of patients?

Chiral separation of D-and L-lactate is of interest in patients in whom plasma lactate (as measured by L-specific lactate dehydrogenase-based assay) is normal, and urinary organic acid analysis shows increased lactate (the sum of L- and D-lactate). Furthermore, increased concentrations of 2-hydroxyisovaleric and 2-hydroxyisocaproic acids may point to D-lactate dehydrogenase deficiency, warranting chiral separation.

1. Surgery, A.S.f.M.a.B. Estimate of Bariatric Surgery Numbers, 2011 - 2017. Vol. 2018 (2018).
2. Borisenko, O. *et al.* Clinical Indications, Utilization, and Funding of Bariatric Surgery in Europe. *Obesity Surgery* **25**, 1408-1416 (2015).
3. Hingorani M, M.A. Aniridia. in *GeneReviews*[®] [Internet] Vol. 2018 (ed. Adam MP, A.H., Pagon RA, et al.) (University of Washington, Seattle, Seattle (WA), 2003).
4. Hanson, I.M. *et al.* PAX6 mutations in aniridia. *Hum Mol Genet* **2**, 915-20 (1993).
5. Takamiya, M. *et al.* Molecular description of eye defects in the zebrafish Pax6b mutant, sunrise, reveals a Pax6b-dependent genetic network in the developing anterior chamber. *PLoS One* **10**, e0117645 (2015).
6. Kleinjan, D.A. *et al.* Subfunctionalization of duplicated zebrafish pax6 genes by cis-regulatory divergence. *PLoS Genet* **4**, e29 (2008).
7. Uribarri, J., Oh, M.S. & Carroll, H.J. D-lactic acidosis. A review of clinical presentation, biochemical features, and pathophysiologic mechanisms. *Medicine (Baltimore)* **77**, 73-82 (1998).
8. Pena, I.A. *et al.* Pyridoxine-Dependent Epilepsy in Zebrafish Caused by Aldh7a1 Deficiency. *Genetics* **207**, 1501-1518 (2017).
9. Reid, E.S. *et al.* Advantages and pitfalls of an extended gene panel for investigating complex neurometabolic phenotypes. *Brain* (2016).
10. Tarailo-Graovac, M., Wasserman, W.W. & Van Karnebeek, C.D. Impact of next-generation sequencing on diagnosis and management of neurometabolic disorders: current advances and future perspectives. *Expert Rev Mol Diagn* **17**, 307-309 (2017).

Reviewer #2 (Remarks to the Author):

This interesting study on detailing the cause of lactate dehydrogenase deficiency has vastly improved in this revised version. The authors made great efforts to respond to all reviewers' comments. I have no further criticisms or additional concerns.

Reviewer #3 (Remarks to the Author):

The authors have addressed the reviewers' comments and questions in a sufficient fashion; they have revised the manuscript accordingly and this certainly resulted in an improved report.

Two minor comments remain:

1) West syndrome is not an etiological diagnosis, ie it is a clinical syndrome describing a constellation of findings. This although I agree it is unclear whether human D lactate dehydrogenase deficiency contributes to the neurological phenotype, they cannot postulate that West syndrome is the cause. Thus the cause of neurologic abn in this patients remains unknown. Please amend.

2) The authors have added that WES and deep phenotyping are important to elucidate dual diagnoses in blended phenotypes. However this still does not cover it; ie the point that CNV (which can be missed by WES) should also be looked for. Would it not be better to reformulate and mention comprehensive genomic analysis is important to screen for CNVs, exonic and cryptic variants, complex rearrangements such as STR expansions. Currently we use a combination of WES and CMA but in the future this may well be WGS only.